# Study of susceptibility to antibiotics and molecular characterization of high virulence *Staphylococcus aureus* strains isolated from a rural hospital in Ethiopia

Cristina Verdú-Expósito[1]*, Juan Romanyk[2], Juan Cuadros-González[2], Abraham TesfaMariam[3], José Luis Copa-Patiño[1], Jorge Pérez-Serrano[1], Juan Soliveri[1]

1 Department of Biomedicine and Biotechnology, University of Alcalá, Alcalá de Henares, Madrid, Spain, 2 Microbiology Service, Hospital Universitario Príncipe de Asturias, Alcalá-Meco, Alcalá de Henares, Madrid, Spain, 3 Department of General Medicine, Gambo General Rural Hospital, West-Arsi, Ethiopia

* cristina.verdu@uah.es

## Abstract

We characterised 80 *Staphylococcus aureus* strains isolated from human patients with SSTIs at a rural hospital in Ethiopia. Susceptibility to antibiotic of all strains was tested. The MLST method was used to type and a phylogenetic analysis was conducted employing the sequences of 7 housekeeping genes. PCR amplification was used to investigate the presence of the following virulence genes in all strains: *hla* (α-haemolysin), *tstH* (toxic shock syndrome toxin), *luk PV* (Panton-Valentine leukocidin), *fnbA* (fibronectin binding protein A) and *mecA* (methicillin resistance). Most of the strains were resistant to penicillin and ampicillin, but only 3 strains were resistant to oxacillin, and 1 of them was a true MRSA. The MLST results showed a high diversity of sequence types (ST), 55% of which were new, and ST152 was the most prevalent. A phylogeny study showed that many of the new STs were phylogenetically related to other previously described STs, but bore little relationship to the only ST from Ethiopia described in the database. Virulence gene detection showed a high prevalence of strains encoding the *hla*, *fnbA* and *pvl* genes (98.77%, 96.3% and 72.84%, respectively), a low prevalence of the *tst* gene (13.58%) and a markedly low prevalence of MRSA (1.25%). *S. aureus* strains isolated from patients in a rural area in Ethiopia showed low levels of antibiotic resistance, except to penicillin. Moreover, this study reveals new STs in Eastern Africa that are phylogenetically related to other previously described STs, and confirm the high prevalence of the *pvl* gene and the low prevalence of MRSA on the continent.

## Introduction

*Staphylococcus aureus* is a gram-positive bacterium with carrier rates of 25–50% in the general population as a commensal microorganism, but it can also become an opportunistic pathogen under certain circumstances. Consequently, it is not only the cause of community-acquired infection (CAI), but is also one of the most important aetiological agents of hospital-acquired

**Data Availability Statement:** All relevant data are within the manuscript and its Supporting Information files.

**Funding:** The author(s) received no specific funding for this work.

**Competing interests:** The authors have declared that no competing interests exist.

infection (HAI) [1]. *S. aureus* presents high plasticity, conferring an exceptional capacity to incorporate genetic material from other strains, and acquire new infection characteristics related to antibiotic resistance and virulence [2]. Hence, phylogenetic studies are very useful to determine the relationship and evolution of different strains, even from different parts of the world [3].

As a result of its versatility and adaptability in the antibiotic era, *S. aureus* has acquired resistance to most of the antibiotics used to treat it [4]. The clearest example of this are the methicillin-resistant *Staphylococcus aureus* strains (MRSA), which express the chromosomal gene *mecA* that encodes a PBP2a transpeptidase, which shows a reduced affinity for all available beta-lactam agents, including penicillin [5, 6]. This gene is located in a small mobile genomic element known as the staphylococcal cassette chromosome (SCC), and is thought to be acquired through horizontal transfer from coagulase-negative staphylococci [7]. MRSA is responsible for a large number of serious infections and hospital-related deaths in developed countries [8, 9], but are also starting to be detected in developing countries [10, 11].

*S. aureus* can also produce a variety of thermostable extracellular protein toxins which behave as virulence factors, including superantigens, haemolysins, leukocidins. The most important of them is the Panton-Valentine leukocidin (PVL), which is a cytotoxin that forms pores in the membrane, and has been associated with furuncles, cutaneous abscesses and severe necrotic skin infections, increasing disease severity [12, 13]. This toxin requires the assembly of two polypeptides, LukS-PV and LukF-PV, into a hetero-oligomeric pore [14]. Although PVL has been related to both community-acquired (CA) MSSA and MRSA (CA-MSSA and CA-MRSA, respectively), an exceptionally large number of MSSAs which express PVL has been isolated in Africa; therefore, this continent has been considered an endemic region for PVL-positive *S. aureus* strains [15].

As noted above, haemolysins are also an important virulence factor. For example, α-haemolysin (Hla) is a pore-forming cytotoxin that can lead to cell lysis and death, producing abscesses in different parts of the body [16]. This toxin is also closely associated with skin and soft tissue infections (SSTI), because it plays a major role in epithelial injury during *S. aureus* infection [17]. It is encoded by the *hla* gene, which is located in the core genome [18].

Another important toxin that affects strain virulence is the toxic shock syndrome toxin (TSST), which is encoded by the *tstH* gene [19, 20]. This exotoxin has multiple biological properties, including the capacity to induce fever, hypotension and lethal shock [21]. The main cause of toxic shock syndrome in humans is the TSST-1 protein (toxic shock syndrome toxin 1) [22].

*S. aureus* pathogenesis is essentially related to the expression of surface fibronectin binding proteins (FnBPs). FnBP-A protein is widely found in body fluids, blood clots and extracellular matrices and its main function appears to be related to the capacity for protein-mediated adhesion of most eukaryotic cells, but it also binds *S. aureus* cells and serves as an adhesion substrate for various microorganisms. This represents a crucial step in the colonisation of host tissue and development of infection [23, 24]. FnBP-A toxin is a glycoprotein formed by two similar subunits and is encoded by the gene *fnbA* located in the chromosomal DNA [25].

Although *S. aureus* infections represent a serious pathogen problem worldwide, studies have largely focused on affluent regions and developed countries, and little information is available on developing countries, especially in East Africa, and particularly in Ethiopia [26]. The Gambo General Rural Hospital is located in the Oromia Region (Ethiopia). Between 2014 and 2018, 80 strains identified as *Staphylococcus aureus* were isolated from patients with extensive, deep subcutaneous purulent lesions, mostly community-acquired SSTIs, which caused recurrent infection in some cases. These strains were mostly methicillin-susceptible (CA-MSSA), except for one resistant CA-MRSA. It is not known if this high virulence was due to the effect of socio-economic and nutritional conditions of the indigenous population or to intrinsic or acquired virulence factors.

The general aim of the present study was to characterise the strains isolated from a rural hospital in Ethiopia in order to determine the origin of their virulence. To this end, susceptibility to antibiotic was tested, and strains were typified by MLST and their sequences were used to study their phylogenetic evolution. Then, the presence of several virulence genes was investigated to determine the origin of the strains' virulence.

## Materials and methods

### Strains

Between June 2014 and June 2018, 80 *Staphylococcus aureus* strains were isolated at Gambo General Rural Hospital from different human samples, mainly SSTIs but also from some patients with osteomyelitis, leprosy and pyomiositis. Most of the strains belonged to children (50), but 22 of them were isolated from adults, and age date of 8 of them could not be collected. In general, strains appeared to be highly virulent and all patients presented deep and extensive lesions, mostly ulcers, and recurrent infection in some cases.

Four different *S. aureus subsp. aureus* strains from the Spanish Type Culture Collection (CECT 976, CECT 957, CECT 435 and CECT 4439) were used as reference controls for different experiments.

In this study, the DNA of each strain was extracted using the UltraClean® Microbial DNA Isolation Kit (MoBio Laboratories Inc., 12224–250) according to the manufacturer's instructions, and this DNA was used for all analyses.

Clinical signs, medical histories and patient characteristics were collected to study any possible relationship with strain characteristics. This study was authorised by the Gambo General Rural Hospital management.

### Ethics statement

After asking to the hospital, the study didn't require any ethics statement because no work was developed with human samples. Strains were isolated directly from the patients to plates.

Strains were collected not only for this study, but also for diagnosing of infection. Patient identifying information was collected by medical doctors as part of the routine hospital patient care procedure, and a number was assigned to each patient.

Information arrived at the laboratory with this number after isolating and identifying all strains. Patient consents for collecting their clinical signs, medical histories, and characteristics were obtained during de admission of the hospital as a part of the routine hospital patient care procedure.

### Identification of strains

First, the preliminary identification of the strains was confirmed by MALDI-TOF (matrix-assisted laser desorption/ionisation-time-of-flight) (MALDI Biotyper System, Bruker), and by partial amplification of 16S gene using the 27F and 1492R primers and sequencing, according to the procedure described by Lane (1991). All sequences were compared against each other for similarity using NCBI-BLAST and the nucleotide collection database (http://blast.ncbi.nlm.nih.gov/Blast.cgi).

### Susceptibility to antibiotics

Susceptibility to antibiotics was analysed by the microdilution broth method using an automated MicroScan system and interpreted with EUCAST expert rules. Briefly, strains were inoculated on Trypticase™ Soy Agar II with 5% Sheep Blood plates (Becton Dickinson,

254053) and incubated for 24 h at 37˚C. An isolated colony of each strain was selected for inoculation on a MicroScan®-Pos MIC Panel Type 33 (Beckman Coulter, B1016-173), a multicell panel with different antibiotics and concentrations. Panels were incubated for 20 h at 37˚C in the MicroScan autoScan4 system (Siemens, B1018-280) with the WalkAway-96 system, which measured final absorbance at 450 nm in each well to determine the presence or absence of growth and therefore susceptibility to antibiotics. This system uses strain *S. aureus* ATCC 25923 as control for this panel. The antibiotics tested were as follows (tested concentrations expressed in μg/mL): penicillin (0.12–0.25,8), ampicillin (0.25, 4–8), amoxicillin + clavulanic acid (4/2-8/4), oxacillin (0.25–2), cefoxitin (4), ceftaroline (0.5–1), imipenem (4–8), gentamicin (1–8), tobramycin (1–8), amikacin (8–32), vancomycin (0.25–16), teicoplanin (1–16), daptomycin (1–4), nitrofuorantoin (32–64), ciprofloxacin (1–2), levofloxacin (1–4), moxifloxacin (0.5–1), erythromycin (0.5–4), clindamycin (0.25–2), quinupristin/dalfopristin (1–4), pristinamycin (1–2), tetracycline (1–8), minocycline (1–8), linezolid (1–4), trimethoprim-sulphamethoxazole (2/38-4/76), rifampicin (0.5–2), fosfomycin (32–64), mupirocin (256) and chloramphenicol (8–16).

Moreover, age of patients was taking into account for comparing with the prevalence of resistances found in each strain.

## Strain typing by MLST

MLST analysis was used to typify all strains (Jolley, Bray, & Maiden, 2018). For that purpose, amplification of 7 housekeeping genes (*arc*, *aro*, *glp*, *gmk*, *pta*, *tpi* and *yqi*) was carried out using the PCR conditions described on the *S. aureus* MLST website (S1 Table), and sequences were obtained by the Sanger method using multi-capillary sequencing (ABI PRISM 3130XL, Applied Biosystems). Then, sequences for each gene from each strain were analysed with BioEdit software and compared with the *S. aureus* MLST database (https://pubmlst.org/saureus/). In accordance with this database, one allele was assigned to each sequence, and the combination of 7 alleles comprised a specific sequence type (ST) for each strain.

## Phylogeny study

Firstly, phylogenetic analysis with all the strains and ST that belong to *S. argenteus* and *S. schweitzeri* was performed to discard that strains belong to these species rather than *S. aureus*.

Then, another phylogenetic analysis was realized with each ST determined previously by MLST typing to know their phylogenetic relations.

Both phylogenetic studies were conducted using the sequences obtained from the 7 housekeeping genes employed in the MLST study. First, the 7 sequences were concatenated in order to create a unique sequence for each strain, and then a multiple alignment of all these sequences was performed with the BioEdit software. G-block software (version 0.91b) was used to eliminate highly variable areas of the sequences. Next, the appropriate substitution model was selected using the jModelTest application (version 2.1.10). Lastly, a maximum likelihood tree was generated and analysed using Mega X software.

## Virulence genes detection by PCR

The virulence genes analysed were *hla*, *tst*, *pvl*, *fnbA* and *mecA*, that codifies for α-haemolysin, toxic shock syndrome toxin, Panton Valentine leucocidin, fibronectin binding protein A and methicillin resistance, respectively. The analyses were carried out by partial PCR amplification of the genes in all strains, under the conditions described in S1 Table [27]. To detect amplified genes, PCR products were resolved using electrophoresis in 1.2% agarose gel (D1 Low EEO, Conda, 8018). The reference controls used to detect each virulence gene were: CECT 976 for

*hla*, CECT 957 for *tst*, CECT 435 for *lukS/F-PV*, CECT 976 for *fnbA* and CECT 4439 for *mecA*, all from the Spanish Type Culture Collection (Spanish initials: CECT). In addition, a molecular weight standard (φX174-Hae III digest, Takara Bio Inc., 3405A) was included to determine the size of the amplified fragments.

On other hand, SCC*mec* typing was performed for positive *mecA* strains, using multiplex PCR described previously [28]. Furthermore, the amplified fragment was sequence and compared with Genbank database.

A relationship between MLST STs and the presence of virulence genes was investigated using Canoco software (version 5.12), conducting a constrained redundancy analysis (RDA).

## Results

### Strain identification

Besides MALDI-TOF identification of *Staphylococcus aureus*, the 16S ribosomal sequences obtained from each sample were compared in the database and identification of the samples as *S. aureus* was confirmed with at least 98% identity for all of them.

### Susceptibility to antibiotics

Results for the MicroScan analysis of all strains are shown in Table 1. Resistance to β-lactam antibiotics was: 88.75% to penicillin, 86.25% to ampicillin, 3.75% to oxacillin and 1.25% to cefoxitin and imipenem. No resistance was detected to ceftaroline.

With regard to β-lactam antibiotics, 3 of the total strains showed resistance to oxacillin, but only 1 of them (strain 73) showed resistance to cefoxitin and imipenem, but also to quinolones (ciprofloxacin, levofloxacin and moxifloxacin).

It is important to note that strain 73 showed the most resistance to the antibiotics used in the assay (13 of 29), and was the only strain isolated that showed resistance to cefoxitin,

**Table 1. Results of MicroScan analysis, showing number and percentage of resistance and intermediate susceptibility (in brackets) of *S. aureus* strains to different antibiotics (rows) pen: penicillin, amp: ampicillin, amc: amoxicillin + clavulanic acid, oxa: oxacillin, fox: cefoxitin, cpt: ceftaroline, ipm: imipenem, gen: gentamicin, tob: tobramycin, amk: amikacin, van: vancomycin, tec: teicoplanin, dap: daptomycin, nit: nitrofuorantoin, cip: ciprofloxacin, lvx: levofloxacin, mxf: moxifloxacin, ery: erythromycin, cli: clindamycin, q-d: quinupristin/dalfopristin, prs: pristinamycin, tet: tetracycline, min: minocycline, lzd: linezolid, sxt: trimethoprim-sulpha-methoxazole, rif: rifampicin, fof: fosfomycin, mup: mupirocin, chl: chloramphenicol.**

| Antibiotic | Number of resistant (and intermediate) strains | % of resistant (and intermediate) strains | Antibiotic | Number of resistant (and intermediate) strains | % of resistant (and intermediate) strains |
|---|---|---|---|---|---|
| pen | 71 | 88.75 | lvx | 1 | 1.25 |
| amp | 69 (1) | 86.25 (1.25) | mxf | 1 | 1.25 |
| amc | 3 | 3.75 | ery | 15 | 18.75 |
| oxa | 3 | 3.75 | cli | 23 (9) | 28.75 (11.25) |
| fox | 1 | 1.25 | q-d | 0 | 0 |
| cpt | 0 | 0 | prs | 0 | 0 |
| ipm | 1 | 1.25 | tet | 42 | 52.5 |
| gen | 12 | 15 | min | 5 | 6.25 |
| tob | 14 | 17.5 | lzd | 0 | 0 |
| amk | 15 | 18.75 | sxt | 4 | 5 |
| van | 0 | 0 | rif | 2 | 25 |
| tec | 0 | 0 | fof | 1 | 1.25 |
| dap | 0 | 0 | mup | 2 | 2.5 |
| nit | 1 | 1.25 | chl | 4 | 5 |
| cip | 1 | 1.25 | | | |

imipenem, trimethoprim-sulphamethoxazole and to all the MLS group (macrolides, lincosamides and streptogramines).

Finally, strains that belonged to adult patients presented more number of resistances than children patients, despite the second group was higher.

## Strain typing by MLST

The results of MLST typing indicated high diversity, and according to the database, over half of the strains (N = 44, 55%) presented new STs (N) (S2 Table). Most of the new STs [38] presented new allele combinations that did not correspond to any ST in the database, while for the remaining 6 STs, there was no match in the MLST database for at least one of the 7 sequences studied and they could not be assigned to any described allele. All these new STs were submitted to database, where a new number of ST was assigned for each one. Most of the new STs [34] comprised only 1 strain, but 4 of them grouped various strains: ST N04 (strains 28, 29, 65 and 83), ST N34 (strains 56, 59 and 74), ST N33 (strains 39 and 46) and ST37 (strains 79 and 84).

We also defined a number of new alleles described for each gene: 2 for the *arc*, *aro* and *yqi* genes, 1 for *glp* and *pta*, and 0 for *gmk* and *tpi*.

Among the STs previously defined in the database (45%), 10 only included 1 strain, another two comprised 2 strains (ST5 and ST3224) and ST121 and ST15 comprised 3 and 4 strains, respectively. The sequence type ST152 was the most prevalent (15 strains) in this study, although with a low percentage (18.75%) with respect to all the strains isolated (Table 2).

## Phylogeny study

A phylogenetic study was conducted using concatenated sequences of MLST housekeeping genes for each strain.

First phylogenetic study confirmed that none of the tested strains belonged to *S. argenteus* or *S. schweitzeri*.

About the second phylogenetic study, after aligning all sequences and discarding all hypervariable regions, an analysis was performed with jModelTest software, selecting the general time-reversible model (GTR) and including invariable sites (+I) and rate of variation across sites (+G) as the best substitution model. This model was used to generate a maximum likelihood (ML) tree with Mega software.

In the ML tree (Fig 1), 15 clusters could be differentiated. The largest group comprised 11 ST, six of them were new, but very similar to other already described. One of these groups (ST N30, N31 and N32) appeared external to the others with a different but closely related origin of the tree. ST N8 also appeared in an individual branch, but with a common branch point to the rest of the strains. The reference strain from Ethiopia (ST727) appeared in a group with ST N10, and this group was clearly distant from the rest, with a common branch except for the most external group mentioned previously.

## Virulence gene detection

The results of PCR virulence detection are shown in Table 2. The most prevalent gene was *hla*, which was detected in all strains except number 24 (thus, 98.77% were α-haemolysin positive strains). In contrast, strain number 73 was the only one that presented the *mecA* gene (1.25% of strains) and one of the three strains did not present the *fnbA* gene (together with strains 15 and 72), indicating that 96.3% of strains were positive for fibronectin binding protein A. In addition, a high percentage of strains (72.84%, 59 strains) showed the *pvl* gene. In contrast, the *tst* gene showed a low prevalence with only 11 strains (13.58%).

**Table 2. Virulence gene detection results.** Table shows presence (+) or absence (-) of all virulence genes studied (*hla*, *tst*, *pvl*, *fnbA*, *mecA*) and for each strain (first column).

| | *hla* | *tst* | *pvl* | *fnbA* | *mecA* |
|---|---|---|---|---|---|
| 1 | + | + | + | + | - |
| 2 | + | + | + | + | - |
| 3 | + | + | + | + | - |
| 4 | + | + | + | + | - |
| 5 | + | + | + | + | - |
| 6 | + | + | + | + | - |
| 7 | + | + | + | + | - |
| 8 | + | - | + | + | - |
| 9 | + | - | - | + | - |
| 10 | + | + | + | + | - |
| 11 | + | + | + | + | - |
| 12 | + | - | + | + | - |
| 15 | + | - | + | - | - |
| 16 | + | - | - | + | - |
| 17 | + | - | - | + | - |
| 18 | + | - | - | + | - |
| 19 | + | - | + | + | - |
| 20 | + | - | + | + | - |
| 21 | + | - | + | + | - |
| 22 | + | - | + | + | - |
| 23 | + | - | + | + | - |
| 24 | - | - | - | + | - |
| 25 | + | - | + | + | - |
| 26 | + | - | - | + | - |
| 27 | + | - | + | + | - |
| 28 | + | - | + | + | - |
| 29 | + | - | + | + | - |
| 30 | + | - | + | + | - |
| 31 | + | - | + | + | - |
| 32 | + | - | + | + | - |
| 33 | + | - | + | + | - |
| 34 | + | - | - | + | - |
| 35 | + | - | + | + | - |
| 36 | + | - | + | + | - |
| 37 | + | - | + | + | - |
| 38 | + | - | - | + | - |
| 39 | + | - | - | + | - |
| 40 | + | - | - | + | - |
| 41 | + | - | - | + | - |
| 42 | + | - | + | + | - |
| 43 | + | - | + | + | - |
| 44 | + | - | + | + | - |
| 45 | + | + | - | + | - |
| 46 | + | - | + | + | - |
| 47 | + | - | - | + | - |
| 48 | + | - | + | + | - |

(*Continued*)

**Table 2.** (Continued)

|  | *hla* | *tst* | *pvl* | *fnbA* | *mecA* |
|---|---|---|---|---|---|
| **50** | + | - | + | + | - |
| **51** | + | - | + | + | - |
| **52** | + | - | + | + | - |
| **53** | + | - | + | + | - |
| **54** | + | - | - | + | - |
| **55** | + | - | + | + | - |
| **56** | + | - | + | + | - |
| **57** | + | - | + | + | - |
| **58** | + | - | + | + | - |
| **59** | + | - | + | + | - |
| **61** | + | - | - | + | - |
| **62** | + | + | - | + | - |
| **63** | + | - | - | + | - |
| **64** | + | - | - | + | - |
| **65** | + | - | + | + | - |
| **66** | + | - | + | + | - |
| **67** | + | - | + | + | - |
| **68** | + | - | + | + | - |
| **69** | + | - | - | + | - |
| **70** | + | - | + | + | - |
| **71** | + | - | + | + | - |
| **72** | + | - | + | - | - |
| **73** | + | - | - | - | + |
| **74** | + | - | - | + | - |
| **75** | + | - | - | + | - |
| **76** | + | - | + | + | - |
| **77** | + | - | + | + | - |
| **78** | + | - | + | + | - |
| **79** | + | - | + | + | - |
| **80** | + | - | + | + | - |
| **81** | + | - | + | + | - |
| **82** | + | - | + | + | - |
| **83** | + | - | + | + | - |
| **84** | + | - | + | + | - |

On other hand, results of SCC*mec* typing determined that strain 73 belong to type IVa, according with the expected size and its sequence (S1).

A correlational analysis using Canoco software indicated that in general there was no relationship between the presence of virulence genes and MLST STs, with an explained cumulative fitted variation of 61.75% with 2 axes, especially in new STs, where virulent genes were highly dispersed (S2).

## Discussion

In contrast to the rest of the world, the picture of the spread of *Staphylococcus aureus* infections in Africa is unclear, especially in Ethiopia, where the few studies available were performed in densely populated urban areas [11, 15].

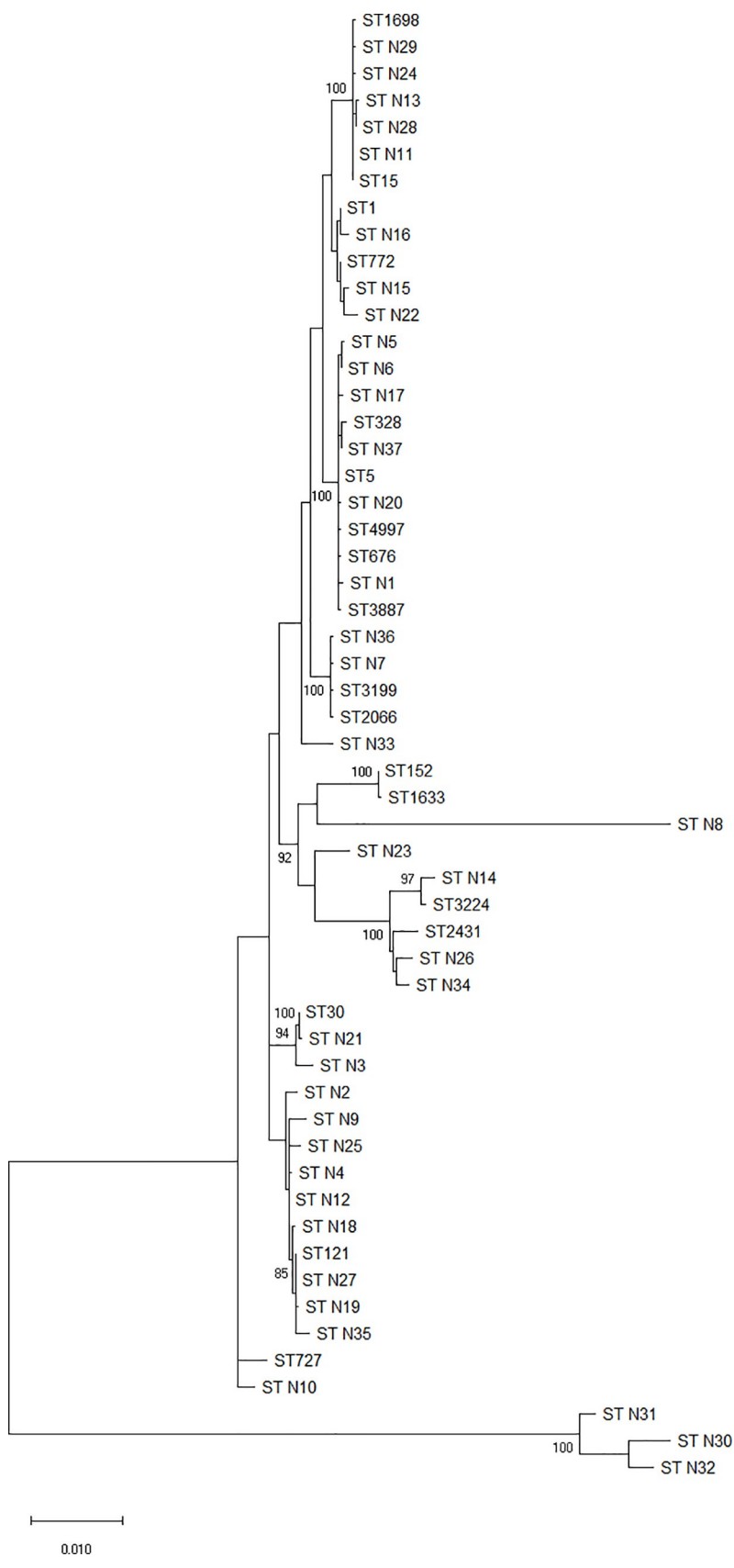

**Fig 1. Maximum likelihood tree.** This phylogenetic diagram was generated using sequences from the 7 housekeeping genes from the MLST analysis, and includes all ST found in this study, but also other STs as reference: the only sequence from the MLST database from Ethiopia (727), the first ST described (ST1), and one of the most common STs found in Africa (ST30), apart from ST152 and ST121 also found in this study.

The study reported here is the first staphylococcal epidemiology study to be conducted in a rural region in Ethiopia, with low population density. Between 2014 and 2018, in Gambo General Rural Hospital, we isolated 80 *S. aureus* strains from clinical samples. All of them caused highly virulent infections with deep and extended lesions, and recurrent infections in some cases.

One of the most important factors in *S. aureus* is the presence of antibiotic resistance. Penicillin was the first and most effective antibiotic against *S. aureus* infections, but also the first resistance emerged [29]. In our study, most of the strains (88%) showed resistance to penicillin and ampicillin, a similar percentage to that found in other studies conducted in Ethiopia [30]. Moreover, 3 of the total isolated strains [38, 43 and 73] exhibited resistance to oxacillin and 1 of them (strain 73) was also resistant to imipenem and cefoxitin. This result suggested that strain 73 was a true MRSA, and this fact was confirmed by amplification and detection of *mecA* gene by PCR, which confers resistance to methicillin (MRSA) and to oxacillin, imipenem and cefoxitin, the latter being a good surrogate marker for MRSA detection. Moreover, type IVa SCC*mec* was determined for strain 73. In addition, MRSA strain 73 was the only strain resistant to quinolones. It has been suggested that fluoroquinolones themselves may predispose patients to infection with MRSA [31]. The percentage of MRSA and resistances to quinolones (ciprofloxacin, levofloxacin or moxifloxacin) obtained in our study was notably lower (1.25%) than that reported in other studies conducted in Ethiopia [32]. This difference may be due to the fact that our study was carried out in a rural area with predominantly community-acquired infections, whereas the other two studies were conducted in a crowded hospital using samples mainly collected from surgical wounds, suggesting hospital-acquired MRSA strains.

Moreover, the collected strains were tested against ceftaroline. All of them were susceptible, probably because it is a fifth-generation cephalosporin that is resistant to *S. aureus* β-lactamase and has a high affinity for PBPs, including the PBP2a present in MRSA [33].

The total percentage of aminoglycoside-resistant strains detected (15–18%) was similar to that found in other studies conducted in Africa [34].

The results for the presence of tetracycline resistance showed that 52.5% of the tested strains were resistant solely to tetracycline and 6.25% to tetracycline and minocycline. These results are similar to those described as average in the African continent [34].

The analysis of resistance to the $MLS_b$ group (macrolides, lincosamides and streptogramine$_b$) determined that 28.75% of strains were only resistant to the lincosamide clindamycin which is very similar that results found in other studies conducted in South Africa (30%) [35]. Resistance to both clindamycin and erythromycin (18%) could be explained by the presence of genes *erm*, which includes resistance to macrolides, lincosamides and streptogramin$_b$ [36]. All strains were susceptible to Sinercyd, a combination of streptogramins A and B, the first of which is not affected by *erm* ribosomal modifications.

In contrast to the high prevalence of trimethoprim-sulphamethoxazole resistance reported in the literature (43–84%), we obtained a low percentage of resistant strains (1.25%) [34]. However, research on *S. aureus* imported from Africa strongly suggests that trimethoprim-sulphamethoxazole resistance is emerging around the globe [37].

The use of antibiotics in a rural area in Ethiopia is more restricted than in the country's large cities and economic centres, and under these conditions, we would expect to find lower

levels antibiotic resistance. Our results confirmed this hypothesis and an analysis of the data showed that resistances were more common in adult patients, who would have been treated more frequently than children with antibiotics. This finding has also been described in urban areas and in developed countries, and in our case was especially notable in strain 73, the most multiple-resistant strain, which was obtained from an adult leprosy patient who had previously been treated repeatedly with different antibiotics.

Sequencing genomes from different *S. aureus* strains revealed that the diversity of genes was high, with 22% belonging to variable regions, rendering the pathogen highly versatile [38]. Hence, surveillance of all *S. aureus* infections around the world is very important to improve our knowledge of their virulence, predict the evolution of strains and infection characteristics and prevent the global spread of multi-resistant or harmful *S. aureus*.

One important tool for surveillance studies is MLST, which uses an online database not only for identification but also to provide knowledge about molecular epidemiology and the global spread of virulent or antibiotic resistant isolates of bacterial pathogens [39]. This analysis has become the method of choice for molecular typing of many bacteria, including *S. aureus*, and for comparing strains around the world [40, 41].

We found a high diversity of STs in general and 44 strains with new STs, suggesting that they did not belong to the same outbreak. This wide diversity has been reported previously in a review of MLST studies conducted in Africa which mainly focused on Central and West Africa. The review identified ST30, ST121 and ST152 as the most prevalent *S. aureus* isolates, all of which were MSSA [15]. We found two of these (ST121 and ST152) in our study, and ST152 was the most prevalent of the two, with 15 strains (18.75%). The MLST database describes 31 isolates of ST152, most of them from Central-West and North Africa, but none of them from Ethiopia. It is noteworthy that the strains identified as ST152 were collected in different years, which could confirm the stability of this ST in this region.

In our study, ST15 and ST121 were also found with 4 and 3 strains, respectively. The MLST database describes 486 isolates of ST15 and 104 isolates of ST121, and both have been found in different parts of the world, including Africa, mainly in the Western region.

Some of the new STs were similar to other, previously described allelic profiles but contained 1 or 2 different alleles: 9 of them were similar to ST121, another 7 to ST5, 6 to ST15 and only 1 to ST152.

The only ST in Ethiopia described in the MLST database is ST727. Hence, ours is the first study in the Eastern region of Africa to identify not only new STs but also STs previously described in other regions of Africa. Moreover, the count of new alleles found in each of the 7 genes showed that the most variable gene in our strains was probably the last one (*yqi*), while *gmk* and *tpi* seemed to be more conserved genes.

Despite the high diversity observed with MLST analysis, phylogenetic studies showing that most of the strains were grouped into clusters and that many of the new STs were phylogenetically related to other, previously described STs. However, a comparison with the only ST described in Ethiopia according to the MLST database demonstrated that our strains were clearly dissimilar (except ST N10) and presented a very different evolution.

The virulence analysis was focused on detecting the main virulence genes of *S. aureus*: *hla*, *tstH*, *lukS/F-PV*, *fnbA* and *mecA*. This analysis indicated a high percentage (72.84%) of PVL-positive strains. One explanation for this high percentage might be that the clinical symptoms were mainly cutaneous (SSTI), because PVL is usually associated with furuncles, cutaneous abscesses and severe necrotic skin infections [12, 42, 43]. However, this high percentage of PVL-positive strains is in agreement with previous studies conducted in other African countries (usually on MSSA), suggesting that there is a high prevalence of *S. aureus* strains presenting genes encoding PVL on this continent [44, 45, 46]. It is considered an endemicity that

could become a reservoir for the emergence of PVL-positive strains. Moreover, it is important to note that in our study, we found a relationship between ST152 and the presence of PVL. This relationship was described in the first studies with a CA-MRSA as a sporadic PVL producer, mostly located in Central Europe. However, later studies in Nigeria and Mali with PVL-positive MSSA ST152 isolates have suggested that ST152 divergence might be the result of a MSSA clone originating in Africa that migrated to Central Europe and acquired antibiotic resistance there, similar to MRSA [46]. This could explain the prevalence of these infections in travellers from Africa, and the strong relationship between ST152 and PVL producers [47, 48].

The *hla* gene was detected in all strains except one [24]. The high prevalence (98.75%) of this gene was similar to that found in other studies on *S. aureus* infections in Africa [49, 50] and throughout the world [51]. It is also important to note that strain 24 was also the most phylogenetically differentiated from the rest.

Only three of the studied strains did not show the *fnba* gene [15, 24 and 72]. This is a high percentage (96.3%) of fnba-encoding genes but is in agreement with previous studies conducted in other African countries [52, 53]. The high prevalence of α-haemolysin and fibronectin binding protein A is not unusual and could explain the high virulence of these infections, as they play a crucial role in colonisation and infection [16, 24].

We found a low prevalence (13.58%) of strains encoding the *tst* gene, which is similar to that found in another study in Congo (17.5%) [54] and to the prevalence of carriers and clinical isolates in Europe (15–25%) [55, 56]. Although TSST-positive infections were initially associated with menstruation, the origin of staphylococcal TSS is diverse, and the most common foci of infection in non-menstrual cases in developed countries are SSTIs [57, 58]. Moreover, it is important to emphasise that most of the samples (9 out of 11; 82%) were collected in the same year (2014), suggesting that there was a high rate of acquisition of this gene during that period.

Statistical analysis showed that there was no relationship between the presence of virulence genes and MLST STs. This could be explained by differences in the acquisition of virulence genes described previously and variability in MLST sequences, but might also be due to the high diversity of STs found.

To sum up, the present study shows a high diversity of *S. aureus* strains in the same rural region, which enhances our knowledge of this kind of infection in Eastern Africa.

## Conclusions

➢ This is one of the first studies on *Staphylococcus aureus* epidemiology in a rural region in Ethiopia.

➢ The prevalence of resistance, especially of MRSA, was low compared with that reported in studies conducted in other parts of Ethiopia and Africa.

➢ MLST analysis showed high diversity of STs and 44 strains with new STs, most of which (new and previously defined STs) are described for the first time in Ethiopia.

➢ Phylogeny studies determined phylogenetic relationships between the new STs and the previously described STs, but far from the only Ethiopian ST in the database.

➢ Virulence gene detection showed a high prevalence of strains encoding *pvl*, *hla* and *fnba*, a low prevalence of the *tss* gene, and a markedly low prevalence of MRSA.

➢ We found no relationship between STs and the presence of virulence genes, but did observe a relationship between ST152 and the PVL gene, highly defined in other parts of Africa.

## Supporting information

**S1 Fig. Alignment of sequences of fragment obtain from multiplex PCR for SCC*mec* typing in strain 73 (Subject) and AB063172 from database GenBank as reference of subtyping IVa of SCC*mec* typing (Query).** Similarity of 100% between both sequences proves that strain 73 present IVa SCC*mec* subtype (59).
(DOCX)

**S2 Fig. Results of statistical correlational analysis presented virulence genes and MLST STs highly dispersed, showing that there was no relationship between these two factors.**
(DOCX)

**S1 Table. Primers and PCR conditions for amplification of different genes.** The PCR reaction mixture was as follows: 10 μl of commercial preparation NZYTaq II 2× Green Master Mix (NZYTech, MB358), 2 μl of primer mixture, 13 μl of DNA-free sterile water and 2 μl of genomic DNA; the final reaction volume was 25 μl.
(DOCX)

**S2 Table. MLST analysis results.** The table shows alleles assigned to each of the seven MLST housekeeping genes (*arc*, *aro*, *glp*, *gmk*, *pta*, *tpi*, *yqi*) and for each strain (first column), as well as sequence type ("ST" column) assigned according to allele combination. "N" sequence type and alleles are assigned to those which do not correspond to any ST (green) or allele (orange) from the database. New STs were assigned numbers (1 to 37). The only ST in the database characterised in Ethiopia (Et) was included.
(DOCX)

## Acknowledgments

We thank the staff at Gambo General Rural Hospital, especially the laboratory workers, and the Microbiology Service at the Príncipe de Asturias University Hospital. We also thank the other members of the University Group for Health Cooperation (UAH-GUdC16-02).

## Author Contributions

**Conceptualization:** Juan Soliveri.

**Data curation:** Juan Romanyk.

**Formal analysis:** Cristina Verdú-Expósito.

**Investigation:** Cristina Verdú-Expósito, Juan Soliveri.

**Methodology:** Cristina Verdú-Expósito, Juan Romanyk, José Luis Copa-Patiño, Juan Soliveri.

**Project administration:** Juan Cuadros-González, Jorge Pérez-Serrano, Juan Soliveri.

**Software:** José Luis Copa-Patiño.

**Supervision:** Jorge Pérez-Serrano, Juan Soliveri.

**Writing – original draft:** Cristina Verdú-Expósito.

**Writing – review & editing:** Cristina Verdú-Expósito, Juan Romanyk, Juan Cuadros-González, Abraham TesfaMariam, José Luis Copa-Patiño, Jorge Pérez-Serrano, Juan Soliveri.

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
