## [Decision Letter · Decision Letter 0]

30 Dec 2019

PONE-D-19-29715

Study of susceptibility to antibiotics and molecular characterisation of high virulence Staphylococcus aureus strains isolated from a rural hospital in Ethiopia

PLOS ONE

Dear Dr. Verdú-Expósito,

Thank you for submitting your manuscript to PLOS ONE. After careful consideration, we feel that it has merit but does not fully meet PLOS ONE’s publication criteria as it currently stands. Therefore, we invite you to submit a revised version of the manuscript that addresses the points raised during the review process.

We would appreciate receiving your revised manuscript by Feb 13 2020 11:59PM. To enhance the reproducibility of your results, we recommend that if applicable you deposit your laboratory protocols in protocols.io, where a protocol can be assigned its own identifier (DOI) such that it can be cited independently in the future. For instructions see: http://journals.plos.org/plosone/s/submission-guidelines#loc-laboratory-protocols

We look forward to receiving your revised manuscript.

Kind regards,

Rosa del Campo

Academic Editor

PLOS ONE

Journal Requirements:

2. Thank you for including your ethics statement of the ethics waiver and patient consent on the submission details page. Please also include those information in your manuscript.

6. Please include a caption for figure 1.

7. We note you have included a table to which you do not refer in the text of your manuscript. Please ensure that you refer to Table 3 in your text; if accepted, production will need this reference to link the reader to the Table.

8. Please include your tables as part of your main manuscript and remove the individual files. Please note that supplementary tables (should remain/ be uploaded) as separate "supporting information" files.

9. Please include a copy of Table 4 which you refer to in your text on page 9.

10. Thank you for stating the following in the Acknowledgments Section of your manuscript:

This work was partially funded by the University of Alcalá, and we also thank the other members of the University Group for Health Cooperation (UAH-GUdC16-02).

Reviewers' comments:

Reviewer's Responses to Questions

**Comments to the Author**

1. Is the manuscript technically sound, and do the data support the conclusions?

Reviewer #1: Yes

Reviewer #2: Partly

2. Has the statistical analysis been performed appropriately and rigorously? 

Reviewer #1: Yes

Reviewer #2: I Don't Know

3. Have the authors made all data underlying the findings in their manuscript fully available?

Reviewer #1: No

Reviewer #2: Yes

4. Is the manuscript presented in an intelligible fashion and written in standard English?

Reviewer #1: Yes

Reviewer #2: Yes

5. Review Comments to the Author

Reviewer #1: Dear Authors

Thanks for your manuscript. the review of the manuscript has been finished and there are some points about it which you could find in the text. especially in quality control of antibiogram and also about the results .

Best Regards

Reviewer #2: This manuscript seems interesting to me from the point of view that presents the molecular characterization of virulence and antibiotic susceptibility of strains of S. aureus isolated in a rural area of Ethiopia, which contributes to the epidemiology of this pathogen in areas where it does not It has enough information. However, it cannot be accepted without made major corrections, as there are some errors.

The following aspect have to be improved:

1. Eighty strains are included, but it would be important to know how many correspond to pediatric origin and how many to adult patients. It is important because in results the author indicate that the resistance was greater in strains recovered from adult patient

2. Synercid is a trade name, in my opinion Quinupristin/dalfopristin, or quinupristin-dalfopristin should be used

3. In methods the authors indicate that the virulence genes are investigated by PCR, and also the products they codify; however, there are not methodology for the latest and also the result of the are missing

4. In page 9, in virulence gene detection section, line 4 I suggest change "samples" by strains

5. In my opinion, there is much speculation in the discussion, because they explain some resistance by referring to genes that the authors do not look for in their work.

6. In page13, line 10, there is a mistake in the ST number: 2ST15 have to be changed to ST152

7. The table 1 have mistake in the % of resistance, column 6. On the other hand, I suggest that the acronyms for the names of antibiotics be 3 letters and according to international standards (i.e. BSAC). In addition, the meaning of each of them must be informed in the table footer. Put the antibiotic in order by families, i.e. beta-lactas, ahminoglycosides, quinolone, etc.

8. In my opinion, table 2 should be summarized and the complete data reported in additional files available online. The names of the genes must be written in lowercase and italic. What does mean the "N"?

9. How was the results of table 3 obtained? I guess they refer to gene products (proteins). I don't find methodology or results for this. Table 3 is not mentioned in the results

10. In my opinion the first conclusion does not apply

11. Finally, considering that they only have one MRSA strain, you should determine the type of SCCmec

6. PLOS authors have the option to publish the peer review history of their article (what does this mean?). If published, this will include your full peer review and any attached files.

Reviewer #1: No

Reviewer #2: No

---

## [Author Response · Author response to Decision Letter 0]

13 Feb 2020

Dear Reviewers and Editor,

Thank you for all your comments and suggestions. We have considered all your recommendations and we have modified the manuscript according with your indications.

Revisions and changes are marked in yellow, and parts deleted are marked with “strikethrough” in the “Revised Manuscript with Track Changes” document.

The manuscript format has been changed to follow the PLOS ONE’s style requirements.

---

## [Editor Report · Decision Letter 1]

20 Feb 2020

Study of susceptibility to antibiotics and molecular characterisation of high virulence Staphylococcus aureus strains isolated from a rural hospital in Ethiopia

PONE-D-19-29715R1

Dear Dr. Cristina Verdú-Expósito,

We are pleased to inform you that your manuscript has been judged scientifically suitable for publication and will be formally accepted for publication once it complies with all outstanding technical requirements.

With kind regards,

Rosa del Campo

Academic Editor

PLOS ONE

---

## [Editor Report · Acceptance letter]

25 Feb 2020

PONE-D-19-29715R1 

Study of susceptibility to antibiotics and molecular characterization of high virulence *Staphylococcus aureus* strains isolated from a rural hospital in Ethiopia 

Dear Dr. Verdú-Expósito:

I am pleased to inform you that your manuscript has been deemed suitable for publication in PLOS ONE. Congratulations! Your manuscript is now with our production department. 

With kind regards,

on behalf of

Dr. Rosa del Campo 

Academic Editor

PLOS ONE